# Velocity gauge formulation of nonlinear optical response in Floquet quantum systems

**S. Sajad Dabiri[1] and Reza Asgari[2,3]**

**1** Department of Physics, Shahid Beheshti University, 1983969411 Tehran, Iran
**2** Department of Physics, Zhejiang Normal University, Jinhua 321004, China
**3** School of Quantum Physics and Matter, Institute for Research
in Fundamental Sciences (IPM), Tehran 19395-5531, Iran

## Abstract

Using the velocity gauge formalism, we develop a theoretical framework for computing the nonlinear optical responses of time-periodic quantum systems. This approach complements the length gauge formulation and offers distinct advantages in both numerical and analytical treatments, particularly for atomic and solid-state systems with well-defined momentum-space structures. By applying our framework to the Rabi model, we derive numerical solutions in the velocity gauge and compare them with the length gauge, demonstrating full agreement between the two formulations. Our findings reveal rich optical phenomena, including photon-assisted transitions, frequency mixing effects, and emergent Floquet-induced photocurrents that are absent in static systems. We demonstrate that nonlinear responses in Floquet-driven systems exhibit resonances at integer multiples of the driving frequency, providing insights into ultrafast spectroscopy and Floquet engineering of quantum materials. The present formulation establishes a bridge between theoretical models and experimental observations in driven quantum systems, with potential applications in quantum optics, photonics, and next-generation optoelectronic devices.

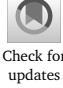

## Contents



# 1 Introduction

Time-periodic quantum system have attracted a lot of attention in last decade due to experimental [1–3] and theoretical [4–6] advancement in studying such systems. Floquet theorem has been proved to be a useful tool to characterize time-periodic systems either in closed non-interacting systems which gives the exact solutions of Schrödinger equation [7–9] or even in time-periodic open quantum systems [10–13] where it can be used to determine the non-equilibrium steady states, i.e non-decaying evolution of these systems. Moreover the Floquet engineering of quantum materials [14–18] and metamaterials [19, 20] has opened a new avenue to induce specific properties on demand.

An intriguing aspect of Floquet systems is their optical response, which emerges from the interplay between the driving field and external probes. The linear response of these systems has been extensively studied [21–29], while understanding their nonlinear responses is crucial for advancing ultrafast photonics and quantum technology. However, a key challenge in describing nonlinear optical effects in Floquet systems is the choice of gauge, which significantly impacts theoretical modeling and computational efficiency. The strong light-matter nonlinear interaction in Floquet systems enables the generation of harmonics at integer multiples of the driving frequency, a process critical for ultrafast light sources and extreme ultraviolet generation.

The study of nonlinear optical responses in static systems has primarily been conducted using two distinct gauges: the velocity gauge, recently formulated with a diagrammatic approach [30], and the length gauge, which has been shown to generate various photocurrents, including shift current, injection current, gyration current, and the intrinsic Fermi surface effect [31].

While they are theoretically equivalent, they can lead to different computational efficiencies and results in practical applications, in particular, in two-band models of graphene and topological insulators due to incomplete basis truncation [32]. In strong field approximation calculations for the ionization of negative ions with odd-parity ground states, the length and velocity gauges produce qualitatively different angular-resolved energy spectra [33], or in the molecular strong field approximation of $N_2$, the length gauge results were found to be in better qualitative agreement with experimental findings compared to the velocity gauge [34]. A key advantage of the velocity gauge is its natural enforcement of optical sum rules, which mitigates truncation errors in numerical simulations. Additionally, the velocity gauge maintains gauge invariance by keeping the Bloch Hamiltonian diagonal in momentum space, facilitating analytical and computational treatments.

The optimal gauge choice depends on the problem, the computational approach, and the physical system under study. Most importantly, it has recently been shown [35] that a careful application of the gauge principle can restore gauge invariance even for extreme light- matter interaction regimes by considering the approximation-induced nonlocality and keeping the resulting interaction Hamiltonian to all orders in the vector potential.

Although our previous work [36] successfully formulated the nonlinear optical response using the length gauge for dynamic systems using the Floquet approach, the complexity of matrix elements in this approach makes numeric treatments challenging. The velocity gauge formulation, in contrast, provides a more straightforward computational framework and ensures gauge-invariant results, making it particularly useful for studying Floquet-driven materials. One advantage of the velocity gauge is that the Bloch Hamiltonian and density matrix perturbation remain diagonal in momentum space. As we will show, the matrix elements in the velocity gauge are straightforward for both analytical approximation and numerical calculations. Moreover, these matrix elements are gauge-invariant, meaning that they do not depend on the specific choice of the phase of the Bloch wave functions across the Brillouin zone.

Since the velocity gauge provides distinct advantages in strong-field physics and Floquet systems, in this work, we develop a comprehensive framework for computing nonlinear optical responses in intrinsically time-periodic quantum systems using the velocity gauge, complementing existing length gauge formulations and providing new insights into Floquet engineering and ultrafast optical phenomena. This formulation provides distinct advantages, particularly for atomic systems and cases that require analytical tractability.

We apply this framework to the Rabi model [37, 38], a paradigmatic example of a driven two-level system, and obtain numerical results. Our results reveal the Rabi model's linear and nonlinear optical responses exhibit complex structures, including photon-assisted transitions and frequency conversion effects that are not present in static systems. Furthermore, we present a numerical solution of the Rabi model in the length gauge. This analysis shows that, while the frequencies and intensity of responses are identical between two gauges at finite frequencies, there is a spurious divergence at zero frequency limit in the velocity gauge which is absent in the length gauge.

Recent advances in ultrafast laser pulses and spectroscopy techniques have enabled probing of nonlinear optical responses with unprecedented precision, as well as Floquet engineering of quantum materials [3, 14, 15, 39–41]. Our work helps bridge the gap between theoretical predictions and experimental observations in driven quantum systems. Therefore, the ability to control nonlinear optical responses in driven Floquet systems provides a new approach to engineering light-driven electronic properties, paving the way for dynamically tunable quantum materials and next-generation optoelectronic devices.

This paper is organized as follows. In Sec. 2, we develop the perturbation theory for calculating the velocity gauge's linear and nonlinear optical responses. Sec. 3 applies this framework to the Rabi model, presenting numerical results for the linear and second-order conductivities in the velocity and length gauge. Sec. 4 concludes the paper with a discussion of the implications of our findings and potential future directions. Detailed derivations and additional results are provided in the Appendices.

## 2 Density matrix and perturbation theory

We consider a solid-state time-periodic system described by the bare Hamiltonian $H(\mathbf{k}, t) = H(\mathbf{k}, t + T)$, where $\mathbf{k}$ is the wave vector and $T = 2\pi/\Omega$ represents the time period. According to Floquet theorem [42] the solutions of Schrödinger equation for this time-periodic Hamiltonian are given by $|\psi_\alpha(t)\rangle = e^{-i\epsilon_\alpha t}|\phi_\alpha(t)\rangle$ where Greek letter $\alpha$ is the band

index, $|\phi_\alpha(t)\rangle = |\phi_\alpha(t + T)\rangle$ is time-periodic Floquet quasimode and $\epsilon_\alpha$ is quasienergy which is defined modulo $\Omega$, i.e. $\epsilon_\alpha \cong \epsilon_\alpha + \Omega$ such that $\epsilon_\alpha$ can be restricted to the first Floquet Brillouin zone i.e. $-\frac{\Omega}{2} < \epsilon_\alpha \le \frac{\Omega}{2}$ (we suppress $\mathbf{k}$ index when it does not make confusion and use natural units $e = \hbar = 1$).

Now, we take a step further and assume that the system is subjected to a probe light. To analyze its response, we develop a perturbation theory. In the velocity gauge, the effect of external light can be incorporated into the Hamiltonian through the Peierls substitution, $\mathbf{k} \to \mathbf{k} + \mathbf{A}(t)$, where $\mathbf{A}(t)$ is the vector potential of the applied probe field.

It is important to note that velocity and length gauges provide the same results; however, due to some practical challenges, one gauge works well for a certain problem rather than the other. The velocity gauge, where the vector potential is used to obtain the electric field, is especially useful in high frequency THz (terahertz) and strong-field physics because it emphasizes kinetic energy contributions. While length gauge may require larger basis sets to capture high-energy transitions accurately. The length gauge is generally simpler when dealing with bound states and many-body systems, like atoms, molecules, or finite systems where dipole moment is a well defined operator [43]. Moreover, the length gauge is more suitable for lower-order perturbation responses as the higher orders involve complicated matrix elements.

The time evolution of the density matrix is derived from the Liouville-von Neumann equation [44, 45]:

$$i\partial_t \rho(\mathbf{k}, t) = \left[ H(\mathbf{k}, t)|_{\mathbf{k} \to \mathbf{k} + \mathbf{A}(t)}, \rho(\mathbf{k}, t) \right], \tag{1}$$

where symbol $[A, B] = AB - BA$ is the commutator. Assuming $\lambda$ as a small constant and $\mathbf{A}(t) = \lambda \mathbf{V}(t)$, we expand $\rho$ in powers of $\lambda$ as $\rho = \rho^{[0]} + \lambda \rho^{[1]} + \lambda^2 \rho^{[2]} + \dots$. Furthermore, we perform a Taylor expansion of the Hamiltonian in terms of powers of $\lambda$ as:

$$H(\mathbf{k}, t)|_{\mathbf{k} \to \mathbf{k} + \mathbf{A}(t)} = h + h^i \lambda V_i(t) + \frac{h^{ij} \lambda^2 V_i(t) V_j(t)}{2!} + \dots, \tag{2}$$

where summation over repeated indices is assumed and $h \equiv H(\mathbf{k}, t)$, $h^i \equiv \partial_{k_i} H(\mathbf{k}, t)$, $h^{ij} \equiv \partial_{k_i} \partial_{k_j} H(\mathbf{k}, t)$.

We assume that the density matrix is diagonal in Floquet basis before applying the perturbation i.e. $\rho^{[0]} = \sum_\eta f_\eta |\phi_\eta(t)\rangle \langle \phi_\eta(t)|$ where the occupation of Floquet states $f_\eta$ being constant in time. The occupation of Floquet states, $f_\eta$, does not follow the Fermi-Dirac distribution function as in static systems. Instead, it can be determined from interactions with photons [24], coupling to the Fermi bath [46], and switch-on protocols [47]. As shown in Appendix A.1, by defining $\rho_{\alpha\beta} \equiv \langle \phi_\alpha(t)|\rho|\phi_\beta(t)\rangle$ and $\epsilon_{\alpha\beta} = \epsilon_\alpha - \epsilon_\beta$, recursion relations between density matrix elements of different orders can be established as follows:

$$\rho_{\alpha\beta}^{[1]} = -i e^{-i\epsilon_{\alpha\beta}t} \int_{-\infty}^{t} dt' e^{i\epsilon_{\alpha\beta}t'} \left[ h^i V_i(t'), \rho^{[0]} \right]_{\alpha\beta},$$

$$\rho_{\alpha\beta}^{[2]} = -i e^{-i\epsilon_{\alpha\beta}t} \int_{-\infty}^{t} dt' e^{i\epsilon_{\alpha\beta}t'} \left( \left[ h^i V_i(t'), \rho^{[1]} \right]_{\alpha\beta} + \left[ \frac{h^{ij}}{2} V_i(t') V_j(t'), \rho^{[0]} \right]_{\alpha\beta} \right)$$

$$= \rho_{\alpha\beta}^{[2]v} + \rho_{\alpha\beta}^{[2]vv}, \tag{3}$$

$$\rho_{\alpha\beta}^{[3]} = -i e^{-i\epsilon_{\alpha\beta}t} \int_{-\infty}^{t} dt' e^{i\epsilon_{\alpha\beta}t'} \left( \left[ h^i V_i(t'), \rho^{[2]} \right]_{\alpha\beta} + \left[ \frac{h^{ij}}{2} V_i(t') V_j(t'), \rho^{[1]} \right]_{\alpha\beta} \right.$$

$$\left. + \left[ \frac{h^{ijl}}{3!} V_i(t') V_j(t') V_l(t'), \rho^{[0]} \right]_{\alpha\beta} \right).$$

## 2.1 First-order optical conductivity

Here, we develop the first-order perturbation of the density matrix as discussed above. Considering $V_z(t) = Ee^{-i\omega_1 t}/i\omega_1$ and substituting this into the first equation of (3), one can determine $\rho_{\alpha\beta}^{[1]}$ (see Appendix A.2).

We find the expectation value of the current $\langle \mathbf{J} \rangle = -\mathcal{N}\langle \mathbf{v}^x \rangle$ (we set the number of charges in the volume $\mathcal{N} = 1$ hereafter and $v^x$ stands for the velocity operator along the probe) and assuming an expansion $\mathbf{J} = \mathbf{J}^{[0]} + \lambda \mathbf{J}^{[1]} + \lambda^2 \mathbf{J}^{[2]} + ...$ and defining $\langle \mathbf{J}^{[1]} \rangle = \sigma^{[1]} E e^{-i\omega_1 t}$. Therefore, we need to find the expectation value of the velocity operator. Note that the velocity operator in the velocity gauge can be obtained by taking the derivative of the total Hamiltonian (2) to the momentum; thus $v^i = h^i + h^{ij}\lambda V_j + \frac{h^{ijk}}{2}\lambda^2 V_j V_k + ... \equiv v^{i[0]} + \lambda v^{i[1]} + \lambda^2 v^{i[2]} + ...$ where we have defined the nth order velocity $v^{i[n]}$. Knowing $\sigma = -\text{Tr}(v^x \rho)/E e^{-i\omega_1 t}$, the first-order conductivity would be obtained by zeroth-order velocity $h^i$ and first-order density matrix $\rho^{[1]}$ or the first-order velocity and zeroth-order density matrix, thus $\sigma^{[1]} = -\text{Tr}(v^{x[1]}\rho^{[0]} + v^{x[0]}\rho^{[1]})/E e^{-i\omega_1 t}$. We find, after straightforward calculations, that the current will have a frequency of $\omega_1 + n\Omega$, $n \in \mathbb{Z}$, so $\sigma^{[1]} = \sum_n e^{-in\Omega t}\sigma^{[1](n)}$ when a probe field with frequency $\omega_1$ is applied to the system. As shown in the Appendix A, we find

$$\sigma_{xz}^{[1](n)}(\omega_1) = \frac{i}{\omega_1}\left(\sum_{\alpha \mathbf{k}} f_\alpha h_{\alpha\alpha}^{xz(n)} - \sum_{\alpha\beta j\mathbf{k}} \frac{f_{\beta\alpha}h_{\beta\alpha}^{x(j+n)}h_{\alpha\beta}^{z(-j)}}{\epsilon_{\alpha\beta} + j\Omega - \omega_1}\right), \tag{4}$$

where the sum over momentum $\mathbf{k}$ indicates the integral over Brillouin zone $\sum_{\mathbf{k}} = \int d^n\mathbf{k}/(2\pi)^n$ and $h_{\beta\alpha}^{x(j)} = 1/T \int_0^T e^{ij\Omega t}\langle \phi_\beta(t)|\partial_{k_x} H(\mathbf{k}, t)|\phi_\alpha(t)\rangle dt$. This formula was first obtained by Kumar et al. [25]. As demonstrated in the Appendix, to determine the optical conductivity in the length gauge, one should replace the matrix elements of the second-order derivative of the Hamiltonian in Eq. (4) with its first-order derivative. The interpretation of Eq. (4) for $n = 0$ is straightforward and involves $j$-photon-assisted optical transitions as shown in Fig. 1(a). The only difference from the static case is the substitution $h_{\beta\alpha} \to h_{\beta\alpha}^{(j)}$ and $\epsilon_{\alpha\beta} \to \epsilon_{\alpha\beta} + j\Omega$. Like as the static case, the divergence in the second term of Eq. (4) at $\omega_1 \to 0$ is spurious and can be canceled by expanding the first term [30, 36, 48]. However, the first term contains a real divergence at $\omega_1 \to 0$ (only when the Floquet bands are partially filled, see Eq. (C.6) of Appendix) associated with the Drude peak, which is well represented in the length gauge.

## 2.2 Second-order optical conductivity

Now, we determine the response of the Floquet system in the second-order of perturbation theory. To achieve this, we need to evaluate $\rho_{\alpha\beta}^{[2]}$ using the second equation of (3), assuming $V_y(t) = E_2 e^{-i\omega_2 t}/i\omega_2$. This evaluation is detailed in Appendix A.3.

Our results show that when electric fields with frequencies of $\omega_1, \omega_2$ are applied to the Floquet system, the second-order response would be at a frequency of $\omega_1 + \omega_2 + n\Omega$, $n \in \mathbb{Z}$, therefore, $\sigma^{[2]} = \sum_n e^{-in\Omega t}\sigma^{[2](n)}$. Defining $\langle \mathbf{J}^{[2]} \rangle = \sigma^{[2]} E E_2 e^{-i(\omega_1 + \omega_2)t}$, the second-order conductivity can be obtained from the zeroth-(second) order velocity matrix and the second-(zeroth) order density matrix and also from the first-order velocity and first-order density matrix as $\sigma^{[2]} = -\text{Tr}(v^{x[2]}\rho^{[0]} + v^{x[1]}\rho^{[1]} + v^{x[0]}\rho^{[2]})/E E_2 e^{-i(\omega_1 + \omega_2)t}$ (see the Appendix A). Since $\rho_{\alpha\beta}^{[2]} = \rho_{\alpha\beta}^{[2]v} + \rho_{\alpha\beta}^{[2]vv}$, the final response of the system at frequency of $\omega = \omega_1 + \omega_2 + n\Omega$

is composed of four contributions as follows:

$$
\sigma_{xyz}^{[2](n)}(\omega_1,\omega_2) = \sum_{\alpha\mathbf{k}} \frac{h_{\alpha\alpha}^{xyz(n)} f_\alpha}{4\omega_1\omega_2} - \frac{1}{2\omega_1\omega_2} \sum_{\alpha\beta j\mathbf{k}} \frac{f_{\beta\alpha} h_{\beta\alpha}^{xy(j+n)} h_{\alpha\beta}^{z(-j)}}{\epsilon_{\alpha\beta} + j\Omega - \omega_1}
$$

$$
- \frac{1}{4\omega_1\omega_2} \sum_{\alpha\beta j_1\mathbf{k}} \frac{f_{\beta\alpha} h_{\beta\alpha}^{x(j_1+n)} h_{\alpha\beta}^{yz(-j_1)}}{\epsilon_{\alpha\beta} + j_1\Omega - \omega_1 - \omega_2}
$$

$$
+ \sum_{\alpha\beta\gamma j_1 j_2\mathbf{k}} \frac{h_{\beta\alpha}^{x(j_1+j_2+n)}/2\omega_1\omega_2}{\epsilon_{\alpha\beta} - \omega_2 - \omega_1 + (j_1+j_2)\Omega} \left\{ \frac{f_{\beta\gamma} h_{\alpha\gamma}^{y(-j_2)} h_{\gamma\beta}^{z(-j_1)}}{\epsilon_{\gamma\beta} - \omega_1 + j_1\Omega} - \frac{f_{\gamma\alpha} h_{\gamma\beta}^{y(-j_1)} h_{\alpha\gamma}^{z(-j_2)}}{\epsilon_{\alpha\gamma} - \omega_1 + j_2\Omega} \right\}
$$

$$
+ (\omega_1, z) \leftrightarrow (\omega_2, y).
$$

(5)

We can establish a simple correspondence between the time-averaged response of Floquet systems $\sigma^{(0)}$ and that of static systems. This correspondence is illustrated in Fig. 1(a), where transitions between the states are now accompanied by $j_1$ or $j_2$ photons. According to Fig. 1(a), two examples of matrix element substitutions are $h_{\beta\gamma} \to h_{\beta\gamma}^{(j_1)}$ and $h_{\beta\alpha} \to h_{\beta\alpha}^{(j_1+j_2)}$. The matrix elements in the reverse direction of the arrows have a minus sign in the Fourier index, i.e., $h_{\gamma\beta} \to h_{\gamma\beta}^{(-j_1)}$. A similar substitution is required for energies according to the direction of the arrows, such that $\epsilon_{\gamma\alpha} \to \epsilon_{\gamma\alpha} - j_2\Omega$ and $\epsilon_{\alpha\gamma} \to \epsilon_{\alpha\gamma} + j_2\Omega$. The summation over Fourier indices like $j_1$ and $j_2$ in Eq. (5) ranges from $(-\ell, \ell)$, with $\ell \to \infty$. In practice, a finite $\ell$ is chosen to ensure the convergence of results. Therefore, $j_1$ and $j_2$ can be replaced with $-j_1$ and $-j_2$ in Eq. (5).

Having utilized this substitution method, we can easily derive higher-order responses of Floquet systems from the formulas used for static systems, which have also diagrammatic representations [30]. The results for third-order conductivity are detailed in Appendix D.

In the next section, we apply our theoretical formalism to calculate the linear and nonlinear optical response of the Rabi model system. For completeness, we extend the model to examine a one-dimensional time-dependent system, with the corresponding results presented in Appendix E.

## 3 Nonlinear optical response of Rabi model

Let us make use of the application of the formalism developed here for a driven two-level atom. The Hamiltonian for this static system in real space is written as

$$
\mathcal{H}(r) = \frac{\Delta}{2}\sigma_z,
$$

(6)

where $\Delta$ is the energy gap between two states and $\sigma_i$, $i \in \{x, y, z, 0\}$ is the Pauli matrix. The position operator in this system is conventionally taken as $\hat{r} = d\sigma_x$, where $d = \langle 1|\hat{r}|2\rangle$ is the element of the off-diagonal matrix of the position operator between the eigenstates of the Hamiltonian. The position operator has no diagonal component since the eigenstates of Hamiltonian are assumed to be the eigenstates of the parity operator. By applying a unitary transformation to Eq. (6) to account all operator orders, we can derive the Bloch Hamiltonian as

$$
\mathcal{H}(k) = e^{-ik\hat{r}}\mathcal{H}(r)e^{ik\hat{r}} = \frac{\Delta}{2}\left(\cos(2dk)\sigma_z - \sin(2dk)\sigma_y\right).
$$

(7)

Note that for a single atom, the quasi-momentum is fixed at a single value, $k = 0$. As mentioned earlier, we apply the Floquet formalism twice, first to the bare dynamic Hamiltonian

and then again when the system is subjected to an external time-dependent probe. In the first step, the bare Hamiltonian can be treated whether in length or velocity gauge. In the length gauge method, where a term $\hat{r} \cdot E(t) = r E_0 \cos \Omega t$ is added to the Hamiltonian, the resulting Hamiltonian becomes

$$\mathcal{H}(k,t) = \frac{\Delta}{2} \left( \cos(2dk)\sigma_z - \sin(2dk)\sigma_y \right) + \Omega_0 \cos \Omega t \, \sigma_x \,, \tag{8}$$

where $\Omega_0 = dE_0$ is the Rabi frequency. However, in the velocity gauge method, which involves using Peierls substitution $k \to k + A(t)$, where $A(t) = -E_0 \sin(\Omega t)/\Omega$, the Hamiltonian is expressed as:

$$\mathcal{H}'(k,t) = \frac{\Delta}{2} \left( \cos(2d(k+A(t)))\sigma_z - \sin(2d(k+A(t)))\sigma_y \right). \tag{9}$$

This represents the correct form of the Hamiltonian in the velocity gauge. Interestingly, Eq. (9) with $k = 0$ was recently proposed in [35] as a solution to the long-standing discrepancy between the length and velocity gauge formulations for systems with a truncated Hilbert space [49]. Equation (9) is equivalent to Eq. (8) and this is provided by moving to a rotating frame using $U_R(t) = \exp[i\hat{r}A(t)] = \exp[iA(t)d\sigma_x]$. Then it is easy to check that

$$\mathcal{H}'(k,t) = U_R^\dagger(t)\mathcal{H}(k,t)U_R(t) - iU_R^\dagger(t) \cdot \partial_t U_R(t). \tag{10}$$

Thus, Eq. (9) can be derived from Eq. (8) via a simple gauge transformation or, equivalently, a change of basis. Crucially, both Eq. (8) and Eq. (9) yield identical optical conductivity. This is because if $|\phi_\alpha(t)\rangle$ is a Floquet quasi-mode of $\mathcal{H}$, then the transformed state $|\phi'_\alpha(t)\rangle = U_R^\dagger(t)|\phi_\alpha(t)\rangle$ becomes a Floquet quasi-mode of $\mathcal{H}'$ with the same quasi-energy. Consequently, the matrix elements appearing in the optical conductivity formulas—Eqs. (4), (5), and (D.1)—are identical for both Hamiltonians. For example:

$$\begin{aligned}
h'^i_{\alpha\beta} &= \langle \phi'_\alpha(t)|\partial_{k_i}\mathcal{H}'(k,t)|\phi'_\beta(t)\rangle \\
&= \langle \phi_\alpha(t)|U_R(t)U_R^\dagger(t)\partial_{k_i}\mathcal{H}(k,t)U_R(t)U_R^\dagger(t)|\phi_\beta(t)\rangle = h^i_{\alpha\beta} \,,
\end{aligned} \tag{11}$$

where we have used the fact that $U_R(t)$ is momentum-independent, which implies $\partial_k[U_R^\dagger(t) \cdot \partial_t U_R(t)] = 0$. In subsequent sections, we adopt Eq. (8) as the time-periodic Hamiltonian for calculating the optical conductivity.

## 3.1 Numerical results in the velocity gauge

For simplicity, we assume the resonant condition $\Delta = \Omega$ and adopt the length gauge for the Hamiltonian given by Eq. (8):

$$\mathcal{H}_{\mathrm{res}}(k,t) = \frac{\Omega}{2} \left( \cos(2dk)\sigma_z - \sin(2dk)\sigma_y \right) + \Omega_0 \cos(\Omega t)\sigma_x \,. \tag{12}$$

In the second step, we want to analyze the nonlinear response of the Hamiltonian (12) and calculate the density matrix given by Eq. (3), hence, we define

$$\begin{aligned}
\mathcal{D} &= \mathcal{H}_{\mathrm{res}}(k,t), \\
\mathcal{D}^x &= \partial_k \mathcal{H}_{\mathrm{res}} = \frac{2d\Omega}{2}\left( -\sin(2dk)\sigma_z - \cos(2dk)\sigma_y \right), \\
\mathcal{D}^{xx} &= \partial_k \partial_k \mathcal{H}_{\mathrm{res}} = \frac{4d^2\Omega}{2}\left( -\cos(2dk)\sigma_z + \sin(2dk)\sigma_y \right), \\
\mathcal{D}^{xxx} &= \partial_k \partial_k \partial_k \mathcal{H}_{\mathrm{res}} = \frac{8d^3\Omega}{2}\left( \sin(2dk)\sigma_z + \cos(2dk)\sigma_y \right).
\end{aligned} \tag{13}$$



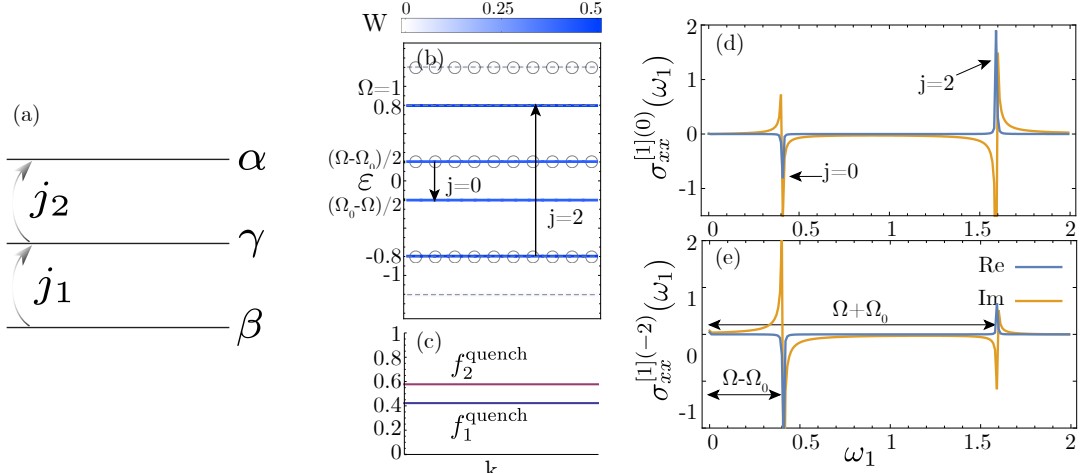

Figure 1: (Color online) (a) Relation between photon-assisted transitions and normal transitions. To find the time average optical response in a Floquet system from the formula for static systems, we should replace the transitions with photon-assisted transitions. The matrix elements should acquire a positive Fourier index along the arrows e.g. $h_{\gamma\alpha} \to h_{\gamma\alpha}^{(j_2)}$ and quasienergies should obtain an additional term in the opposite direction of the arrows. e.g. $\epsilon_{\gamma\beta} \to \epsilon_{\gamma\beta} + j_1\Omega$. (b) quasi-energy bands with the color scale representing the physical weight of the sidebands and (c) quench occupation of the Floquet states. (d), and (e) the first-order optical response of resonant Rabi model to a probe field at a frequency of $\omega_1$ using Eqs. (4), (13). Notice that $j = 0$, and 2 photon transitions are allowed and noticeable peaks occur at $\Omega_0 \pm \Omega$. The circles in panel (b) show that a band is more filled with respect to a band without circles, also the color scale shows the physical weight defined in the main text. The $j$-photon-assisted optical transitions are indicated by arrows in (b). The parameters are $\Omega = 1, \Omega_0 = 0.6, d = 0.5$.

As mentioned previously, the occupation of the Floquet states depends on various factors, such as switch-on protocols and dissipation. We can assume the *quench occupation* of the Floquet states, which occurs when the drive is suddenly turned on after the system has been in the ground state of the static two-level atom. This occupation is determined by projecting the Floquet states onto the ground state $f_\alpha^{\text{quench}} = \sum_n |\langle \phi_\alpha^{(n)}|1\rangle|^2$ where $|\phi_\alpha^{(n)}\rangle = 1/T \int_0^T e^{in\Omega t}|\phi_\alpha(t)\rangle dt$ and for the Hamiltonian (13) and a special set of parameters as illustrated in Fig. 1(b) and (c). It is seen that the lower Floquet band (with a lower quasi-energy in the first Floquet Brillouin zone, i.e. $(-\Omega/2 < \epsilon < \Omega/2)$) has a lower occupation.

The first-order conductivity can be calculated using Eqs. (4) and (13). The finite Fourier components are $n = 0, -2$. Figure 1(d) and (e) presents the results for the first-order conductivity of the Rabi model, calculated for a specific set of parameters. A small phenomenological imaginary constant $\eta \sim 0.003$ can be added to the frequency $\omega_1 \to \omega_1 + i\eta$ to account for the electron relaxation process. Nonetheless, as suggested in Ref. [50], the replacement $\omega_1 \to \sqrt{\omega_1(\omega_1 + i\eta)}$ will give better results at low frequencies in the velocity gauge, however, this may cause fluctuations in optical conductivity which necessitates using finer discretization of Brillouin zone. We take this assumption in the following. The quasi-energy spectrum is shown in Fig. 1(b) with the color scale representing the physical weight of the sidebands, $W_\alpha = \langle \phi_\alpha^{(n)}|\phi_\alpha^{(n)}\rangle$. As seen in Fig. 1(b) the only side bands with finite weights are the first (second) Floquet bands of +1(-1) replicas and also two Floquet bands of the zeroth replicas. All other bands have negligible weight. The positions and signs of the peaks and dips in Figs. 1(d)



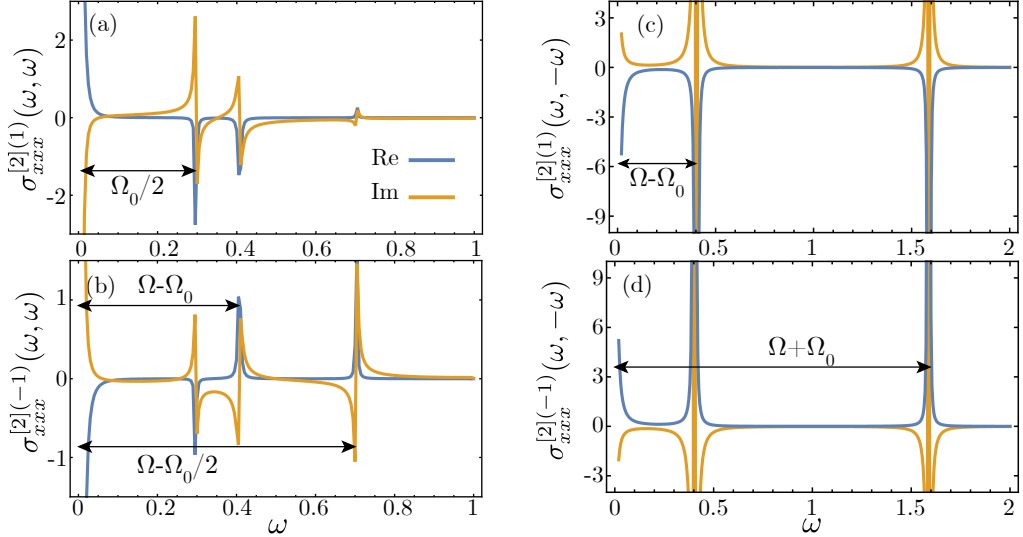

Figure 2: (Color online) Nonzero elements of the second-order optical conductivity of resonant Rabi model calculated from Eqs. (5) and (13). The peaks and dips align with the resonant transitions in the system. It is important to notice that various resonant frequencies occur in each term. The parameters are $\Omega = 1, \Omega_0 = 0.6, d = 0.5$ and $\omega_1 = \omega_2 = \omega$ in (a) and (b) while $\omega_1 = \omega_2 = -\omega$ in (c) and (d).

and (e) correspond to the optical $j$-photon-assisted transitions illustrated in Fig.1(b). Notably, only $j = 0, 2$ photon transitions are allowed due to the special structure of the drive. An effect of the light is to modify the effective gap of the system, shifting it from $\Omega$ to $\Omega - \Omega_0$, as illustrated in Fig. 1(b). Another effect is the introduction of an additional absorption line at the frequency $\Omega + \Omega_0$, which does not exist in the static system.

Let us analyze some special cases based on Fig. 1. If $\Omega_0 = \Omega$ then according to Fig. 1(e) a dc probe field would produce an intense response at the frequency of $-2\Omega$. Conversely, a probe field at a frequency of $2\Omega$ would induce a response at zero frequency, resulting in a rectified current. It is important to note that these frequency conversions occur at the first-order of response, unlike in static systems, where the first-order perturbation theory predicts a response at the same frequency as the probe field.

Now, let us calculate the second-order responses of the resonant Rabi model using Eqs. (5) and (13). First, we consider the case where $\omega_1 = \omega_2 = \omega$. The second-order optical conductivity in Eq. (5) is finite only for $n = \pm 1$ and consists of four contributions. The total results are plotted in Figs. 2(a) and (b) as a function of probe frequency. The peaks and dips align with the resonant transitions in the system. Figure 2 illustrates various nonlinear phenomena. For example, consider the special case where $\Omega = \Omega_0$. According to Figs. 2(a) and (b), a dc probe field will cause intense responses at frequencies $\pm\Omega$. Moreover, a probe field at the frequency of $\Omega/2$ will induce responses at frequencies $3\Omega/2$ and $-\Omega/2$. These diverse optical phenomena can be tested in experimental setups, especially because of the discrete energy levels leading to novel frequency conversion effects.

Next, we consider the case where $\omega_1 = -\omega_2 = \omega$. The calculations reveal that the only non-negligible Fourier components for optical conductivity in this scenario are $n = \pm 1$. Using Eqs. (5) and (13), it is straightforward to calculate four contributions as shown in Figs. 2(c) and (d). Notice that we introduced a small constant $\omega_1 + \omega_2 \approx 3\eta$ to avoid the singularity in the denominator of the last equation of (5). We observe that the only resonant frequencies in this case occur at $\omega = \Omega + \Omega_0$ and $\omega = \Omega - \Omega_0$. As a limiting case, if we consider $\Omega = \Omega_0$, then

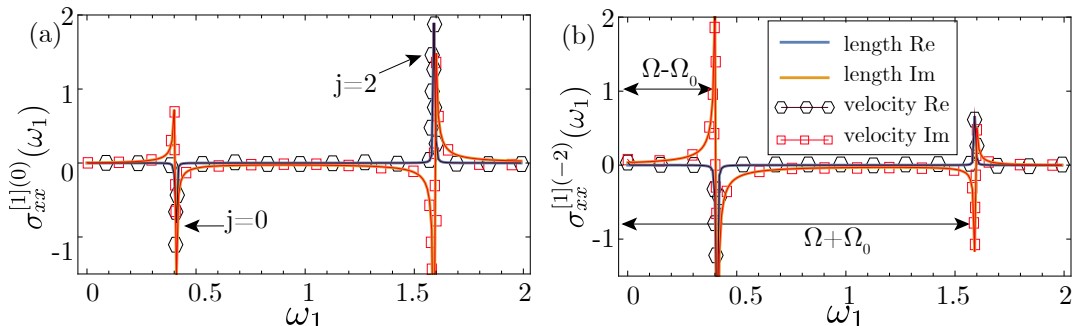

Figure 3: (Color online) (b),(c) The first-order optical response of resonant Rabi model to a probe field at a frequency of $\omega_1$ in the length gauge (solid line) and velocity gauge (hexagons and squares), respectively. Notice that $j = 0$, and 2 photon-assisted transitions are active. The parameters are $\Omega = 1, \Omega_0 = 0.6, d = 0.5$.

according to Figs. 2(c) and (d), a probe field at a frequency of $\omega = 0$ would induce significant currents at frequencies of $\pm\Omega$. Furthermore, a probe field at a frequency of $\omega = 2\Omega$ would cause responses at frequencies of $\Omega$ and $3\Omega$.

The nonlinear response of the Rabi model using the length gauge formalism [36] is presented in the next section. The results exhibit many similarities, especially away from the zero frequency limit ($\omega = 0$), the nonlinear conductivities are the same. Consequently, the frequencies of the peaks and dips remain identical. Nevertheless, there is a spurious divergence at the zero-frequency limit in the velocity gauge formalism, which is absent in the length gauge formulation.

## 3.2 Numerical results in the length gauge

For the sake of completeness, let us analyze the nonlinear response of the resonant Rabi model in the length gauge using numerical methods. The Hamiltonian and its derivatives are provided in Eq. (13). As outlined in Eq. (A.2) in the Appendix, the time-dependent Schrödinger equation can be transformed into a time-independent eigenvalue problem with an infinite-dimensional static Hamiltonian by employing the Fourier decomposition of quasi-modes of Floquet and the Hamiltonian. Fortunately, this Hamiltonian can be truncated to obtain approximate solutions.

We truncate the time-independent Hamiltonian (A.2) to include Floquet replicas in the range $n \in [-3, 3]$. The linear and second-order response formulas in the length gauge are provided in Eq. (5), (6) and Eq.(8), (c2) of Ref. ([36]), respectively. In particular, the position operator is taken as $\hat{r} = d\sigma_x$ and the velocity operator in the length gauge ($\mathcal{D}^x$) is represented in Eq. (13).

Figures 3(a) and (b) show the first-order response under the assumption of quench occupation. Here, again, only $j = 0, 2$ photon-assisted transitions are active, giving rise to the two spikes observed in Figs. 3(a) and (b) which are consistent with the quasienergy band structure, physical weights, and dynamical gap in Fig. 1(b). The limit of zero frequency also shows no optical conductivity, which is justified by the flat band structure. The results of the velocity gauge are also shown in Fig. 3 marked with hexagons and squares showing agreement between the two gauges.

The second-order responses for the cases $\omega_1 = \omega_2 = \omega$ and $\omega_1 = -\omega_2 = \omega$ are plotted in Figs. 4(a), (b), and (c), (d), respectively. The velocity gauge results are also plotted for comparison. The length gauges results and velocity gauges results are in complete agreement except for the zero-frequency limit. Indeed, the numerical results in the length gauge are more reliable. However, a smooth gauge must be chosen for the phase of the Bloch wave function across the Brillouin zone to achieve the correct form.

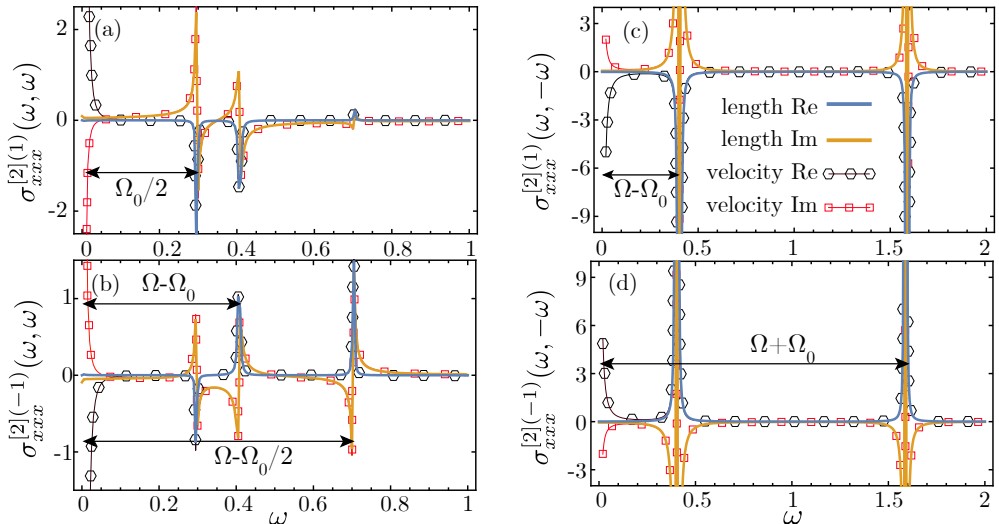

Figure 4: (Color online) Nonzero elements of the second-order optical conductivity of the resonant Rabi model in length (solid line) and velocity (hexagons and squares) gauges, respectively. The length gauges results and velocity gauges results are in complete agreement except for the zero-frequency limit. The parameters are $\Omega = 1, \Omega_0 = 0.6, d = 0.5$ and $\omega_1 = \omega_2 = \omega$ in (a) and (b) while $\omega_1 = \omega_2 = -\omega$ in (c) and (d).

## 4 Conclusion

We developed a framework for calculating the linear and nonlinear optical responses of intrinsically time-periodic systems using the velocity gauge at zero temperature and in pure systems. The original Hamiltonian is assumed to be time-periodic, while an external field, typically weak, is applied with a different frequency. We investigated the system's nonlinear optical response, which is proportional to the powers of the probe's electric field. We have used the velocity gauge which is suited for Floquet systems since it simplifies the treatment of time-periodic driving fields, ensures gauge invariance, and provides a more efficient and numerically stable framework for computing nonlinear optical responses.

We have applied this framework to a driven two-level atom, known as the Rabi model, and derived numerical results. The length gauge nonlinear response of the Rabi model reveals similar linear and nonlinear responses, with the frequencies of peaks and dips nearly identical to those in the velocity gauge case. Our findings provide a foundation for interpreting the nonlinear optical responses of driven systems, which have recently become experimentally accessible thanks to advances in ultrashort laser pulses and ultrafast spectroscopy techniques [15,39,51].

Our velocity gauge framework offers a robust approach for computing nonlinear optical responses in Floquet-driven materials. We predict that experimental observation of these effects is feasible under the following realistic conditions: Strong-field ultrafast laser sources in the THz to visible range, Low temperatures (10–100 K) to minimize decoherence, High-purity samples with minimal disorder to preserve Floquet-band coherence, Pump-probe spectroscopy and nonlinear optical measurements to capture Floquet-induced currents. These conditions are well within the reach of modern condensed matter laboratories, making our predictions testable with current experimental capabilities.

Although our formalism provides accurate results for a two-band topological system, it can be extended to a broader range of cases, including the application of the velocity gauge formulation in multiband systems and strongly correlated electron materials.

# A    Details of derivations

In this section, we provide detailed calculations, specifically demonstrating how density matrices and conductivities of various orders are obtained.

The fundamentals of Floquet theory are extensively studied in the literature. The Schrödinger equation for Floquet quasi-modes, as defined in the main text, can be represented as:

$$(H(t) - i\partial_t)|\phi_\alpha(t)\rangle = \epsilon_\alpha|\phi_\alpha(t)\rangle, \tag{A.1}$$

where the momentum index $\mathbf{k}$ is suppressed. If we insert the Fourier expansion of time-periodic Hamiltonian and Floquet quasi modes $H(t) = e^{-in\Omega t}H^{(n)}$ and $|\phi_\alpha(t)\rangle = e^{-im\Omega t}|\phi_\alpha^{(m)}\rangle$ in (A.1) we find

$$\sum_n H^{(m-n)}|\phi_\alpha^{(n)}\rangle - m\Omega|\phi_\alpha^{(m)}\rangle = \epsilon_\alpha|\phi_\alpha^{(m)}\rangle. \tag{A.2}$$

This equation is an eigenvalues problem with a static infinite dimensional Hamiltonian. This Hamiltonian in the numerical calculations should be truncated at a dimension that guarantees the convergence of the results (also see [36]). It can also be shown that an orthogonality relation between Floquet quasi-modes can be defined as

$$\langle\phi_\alpha(t)|\phi_\beta(t)\rangle = \delta_{\alpha\beta}, \tag{A.3}$$

which we assume hereafter.

## A.1    Perturbation theory

To derive Eq. (3) of the main text we substitute the expansion of $\rho$ and also total Hamiltonian (Eq. (2)) in powers of $\lambda$ in the von Neumann equation and equate the terms with the same powers of $\lambda$ and find

$$i\partial_t\rho^{[0]} = [h, \rho^{[0]}],$$
$$i\partial_t\rho^{[1]} = [h, \rho^{[1]}] + [h^i V_i(t), \rho^{[0]}],$$
$$i\partial_t\rho^{[2]} = [h, \rho^{[2]}] + [h^i V_i(t), \rho^{[1]}] + \left[\frac{h^{ij}}{2!}V_i(t)V_j(t), \rho^{[0]}\right], \tag{A.4}$$
$$i\partial_t\rho^{[3]} = [h, \rho^{[3]}] + [h^i V_i(t), \rho^{[2]}] + \left[\frac{h^{ij}}{2!}V_i(t)V_j(t), \rho^{[1]}\right] + \left[\frac{h^{ijl}}{3!}V_i(t)V_j(t)V_l(t), \rho^{[0]}\right].$$

We can find the matrix elements in Eq. (A.4) by using Eq. (A.1) as follows

$$\langle\phi_\alpha(t)|[h, \rho^{[n]}]|\phi_\beta(t)\rangle = \epsilon_{\alpha\beta}\langle\phi_\alpha(t)|\rho^{[n]}|\phi_\beta(t)\rangle - i\langle\partial_t\phi_\alpha(t)|\rho^{[n]}|\phi_\beta(t)\rangle$$
$$- i\langle\phi_\alpha(t)|\rho^{[n]}|\partial_t\phi_\beta(t)\rangle. \tag{A.5}$$

Substituting Eq. (A.5) in Eq. (A.4) we reach at

$$i\partial_t\rho_{\alpha\beta}^{[0]} = (\epsilon_\alpha - \epsilon_\beta)\rho_{\alpha\beta}^{[0]} \equiv \epsilon_{\alpha\beta}\rho_{\alpha\beta}^{[0]},$$
$$i\partial_t\rho_{\alpha\beta}^{[1]} = \epsilon_{\alpha\beta}\rho_{\alpha\beta}^{[1]} + [h^i V_i(t), \rho^{[0]}]_{\alpha\beta},$$
$$i\partial_t\rho_{\alpha\beta}^{[2]} = [h^i V_i(t), \rho^{[1]}]_{\alpha\beta} + \left[\frac{h^{ij}}{2}V_i(t)V_j(t), \rho^{[0]}\right]_{\alpha\beta}, \tag{A.6}$$
$$i\partial_t\rho_{\alpha\beta}^{[3]} = [h^i V_i(t), \rho^{[2]}]_{\alpha\beta} + \left[\frac{h^{ij}}{2}V_i(t)V_j(t), \rho^{[1]}\right]_{\alpha\beta} + \left[\frac{h^{ijl}}{2}V_i(t)V_j(t)V_l(t), \rho^{[0]}\right]_{\alpha\beta}.$$

By changing the variables $\rho_{\alpha\beta}^{[n]} = S(t)e^{-i\epsilon_{\alpha\beta}t}$ therefore $i\partial_t\rho_{\alpha\beta}^{[n]} = i\partial_t S(t)e^{-i\epsilon_{\alpha\beta}t} + \epsilon_{\alpha\beta}S(t)e^{-i\epsilon_{\alpha\beta}t}$ we can integrate the equations for $S(t)$ in (A.6) and prove Eq. (3).

## A.2 First-order conductivity

Now, we find the first-order density matrix. By inserting $V_z(t) = Ee^{-i\omega_1 t}/i\omega_1$ in the first equation of (3) and using the fact that $\rho^{[0]}_{\alpha\beta} = f_\alpha \delta_{\alpha\beta}$ we find:

$$
\begin{aligned}
\rho^{[1]}_{\alpha\beta} &= -\frac{ie^{-i\epsilon_{\alpha\beta}t}E}{i\omega_1} \sum_\gamma \int_{-\infty}^t e^{i(\epsilon_{\alpha\beta}-\omega_1)t'} \left( h^z_{\alpha\gamma}\rho^{[0]}_{\gamma\beta} - \rho^{[0]}_{\alpha\gamma}h^z_{\gamma\beta} \right) dt' \\
&= -\frac{ie^{-i\epsilon_{\alpha\beta}t}E}{i\omega_1} \int_{-\infty}^t e^{i(\epsilon_{\alpha\beta}-\omega_1)t'} \left( f_\beta - f_\alpha \right) h^z_{\alpha\beta} dt', \\
\rho^{[1]}_{\alpha\beta} &= \frac{iE}{\omega_1} \sum_j \frac{e^{i(-\omega_1+j\Omega)t}}{\epsilon_{\alpha\beta} + j\Omega - \omega_1} f_{\beta\alpha} h^{z(-j)}_{\alpha\beta},
\end{aligned}
\tag{A.7}
$$

where $h^{z(-j)}_{\alpha\beta} = 1/T \int_0^T e^{-ij\Omega t} \langle \phi_\alpha(t) | \partial_{k_z} h | \phi_\beta(t) \rangle dt$.

The first-order conductivity can be determined by using the zeroth-order velocity and first-order density matrix or first-order velocity and zeroth-order density matrix as

$$
\begin{aligned}
\sigma^{[1]} &= -\text{Tr}\left( \frac{v^{x[1]}\rho^{[0]} + v^{x[0]}\rho^{[1]}}{Ee^{-i\omega_1 t}} \right) = \frac{-\sum_\alpha h^{xz}_{\alpha\alpha}V(t)\rho^{[0]}_{\alpha\alpha} - \sum_{\alpha\beta} h^x_{\beta\alpha}\rho^{[1]}_{\alpha\beta}}{Ee^{-i\omega_1 t}} \\
&= \frac{i}{\omega_1}\left( \sum_\alpha h^{xz}_{\alpha\alpha}f_\alpha - \sum_{\alpha\beta j} \frac{e^{ij\Omega t}h^x_{\beta\alpha}f_{\beta\alpha}h^{z(-j)}_{\alpha\beta}}{\epsilon_{\alpha\beta} + j\Omega - \omega_1} \right), \\
\sigma^{[1](n)}_{xz}(\omega_1) &= \frac{i}{\omega_1}\left( \sum_\alpha f_\alpha h^{xz(n)}_{\alpha\alpha} - \sum_{\alpha\beta j} \frac{h^{x(j+n)}_{\beta\alpha}f_{\beta\alpha}h^{z(-j)}_{\alpha\beta}}{\epsilon_{\alpha\beta} + j\Omega - \omega_1} \right).
\end{aligned}
\tag{A.8}
$$

## A.3 Second-order conductivity

We evaluate the second-order density matrix by using the second equation of (3) and Eq. (A.7). By defining $V_y(t) = E_2 e^{-i\omega_2 t}/i\omega_2$ we have

$$
\rho^{[2]v}_{\alpha\beta} = -\frac{ie^{-i\epsilon_{\alpha\beta}t}E_2}{i\omega_2} \sum_\gamma \int_{-\infty}^t e^{i(\epsilon_{\alpha\beta}-\omega_2)t'} \times \left( h^y_{\alpha\gamma}\rho^{[1]}_{\gamma\beta} - \rho^{[1]}_{\alpha\gamma}h^y_{\gamma\beta} \right) dt',
$$

$$
\rho^{[2]v}_{\alpha\beta} = -\frac{EE_2}{\omega_1\omega_2} \sum_{j_1 j_2 \gamma} \frac{e^{i(-\omega_1-\omega_2+(j_1+j_2)\Omega)t}}{\epsilon_{\alpha\beta} + (j_1+j_2)\Omega - \omega_1 - \omega_2} \left[ \frac{f_{\beta\gamma}h^{y(-j_2)}_{\alpha\gamma}h^{z(-j_1)}_{\gamma\beta}}{\epsilon_{\gamma\beta} + j_1\Omega - \omega_1} - \frac{f_{\gamma\alpha}h^{z(-j_1)}_{\alpha\gamma}h^{y(-j_2)}_{\gamma\beta}}{\epsilon_{\alpha\gamma} + j_1\Omega - \omega_1} \right], \tag{A.9}
$$

$$
\rho^{[2]vv}_{\alpha\beta} = -\frac{ie^{-i\epsilon_{\alpha\beta}t}EE_2}{2i\omega_2 i\omega_1} \sum_\gamma \int_{-\infty}^t e^{i(\epsilon_{\alpha\beta}-\omega_1-\omega_2)t'} \left( h^{yz}_{\alpha\gamma}\rho^{[0]}_{\gamma\beta} - \rho^{[0]}_{\alpha\gamma}h^{yz}_{\gamma\beta} \right) dt',
$$

$$
\rho^{[2]vv}_{\alpha\beta} = \frac{EE_2}{2\omega_1\omega_2} \sum_{j_1\gamma} \frac{e^{i(-\omega_1-\omega_2+j_1\Omega)t}}{\epsilon_{\alpha\beta} + j_1\Omega - \omega_1 - \omega_2} \left[ f_{\beta\alpha}h^{yz(-j_1)}_{\alpha\beta} \right]. \tag{A.10}
$$

Note that Eq. (A.9) can be written in an equivalent form by changing the variables in the second term $j_1 \leftrightarrow j_2$ and also $(z, \omega_1) \leftrightarrow (y, \omega_2)$ which is justified because we later symmetrize the conductivities. Hence,

$$
\rho^{[2]v}_{\alpha\beta} = -\frac{EE_2}{\omega_1\omega_2} \sum_{j_1 j_2 \gamma} e^{i(-\omega_1-\omega_2+(j_1+j_2)\Omega)t} h^{y(-j_2)}_{\alpha\gamma} h^{z(-j_1)}_{\gamma\beta} I(\omega_1, \omega_2), \tag{A.11}
$$

$$
I(\omega_1, \omega_2) = \frac{1}{\epsilon_{\alpha\beta} + (j_1+j_2)\Omega - \omega_1 - \omega_2} \left[ \frac{f_{\beta\gamma}}{\epsilon_{\gamma\beta} + j_1\Omega - \omega_1} - \frac{f_{\gamma\alpha}}{\epsilon_{\alpha\gamma} + j_2\Omega - \omega_2} \right]. \tag{A.12}
$$

Next, we calculate the second-order conductivity using the zeroth (second)-order velocity and the second (zeroth)-order density matrix, as well as the first-order velocity and the first-order density matrix. After symmetrizing the final result with respect to $(\omega_1, z) \leftrightarrow (\omega_2, y)$, we obtain

$$\sigma^{[2]} = -\text{Tr}(v^{x[2]}\rho^{[0]} + v^{x[1]}\rho^{[1]} + v^{x[0]}\rho^{[2]})/EE_2 e^{-i(\omega_1+\omega_2)t}, \tag{A.13}$$

$$\sigma^{[2](n)}(\omega_1, \omega_2) = \frac{\sum\limits_{\alpha\mathbf{k}} h_{\alpha\alpha}^{xyz(n)} f_\alpha}{4\omega_1\omega_2} - \frac{1}{2\omega_1\omega_2} \sum_{\alpha\beta j\mathbf{k}} \frac{f_{\beta\alpha} h_{\beta\alpha}^{xy(j+n)} h_{\alpha\beta}^{z(-j)}}{\epsilon_{\alpha\beta} + j\Omega - \omega_1}$$

$$- \frac{1}{4\omega_1\omega_2} \sum_{\alpha\beta j_1\mathbf{k}} \frac{f_{\beta\alpha} h_{\beta\alpha}^{x(j_1+n)} h_{\alpha\beta}^{yz(-j_1)}}{\epsilon_{\alpha\beta} + j_1\Omega - \omega_1 - \omega_2}$$

$$+ \frac{1}{2\omega_1\omega_2} \sum_{\alpha\beta\gamma j_1 j_2\mathbf{k}} h_{\beta\alpha}^{x(j_1+j_2+n)} h_{\alpha\gamma}^{y(-j_2)} h_{\gamma\beta}^{z(-j_1)} I(\omega_1, \omega_2) + ((\omega_1, z) \leftrightarrow (\omega_2, y)),$$

which is equivalent to the expressions (5) of the main text.

# B Velocity operator in the length gauge and velocity gauge

In this section, we compare the velocity operator in the length and velocity gauges. As stated in the main text, in the length gauge, the electric field is coupled to the position, and the total Hamiltonian is written as

$$H_{\text{tot}}^E(\mathbf{k}, t) = H(\mathbf{k}, t) + \mathbf{r} \cdot \mathbf{E}(t). \tag{B.1}$$

Using the Heisenberg equation of motion, one can express the velocity operator in the length gauge ($\mathbf{v}^E$) as

$$\mathbf{v}^E = \frac{d\mathbf{r}}{dt} = i\left[H_{\text{tot}}^E(\mathbf{k}, t), \mathbf{r}\right] = i\left[H(\mathbf{k}, t) + \mathbf{r} \cdot \mathbf{E}(t), \mathbf{r}\right]$$
$$= i\left[H(\mathbf{k}, t), \mathbf{r}\right] = i\partial_\mathbf{k} H(\mathbf{k}, t)[\mathbf{k}, \mathbf{r}] = \partial_\mathbf{k} H(\mathbf{k}, t), \tag{B.2}$$

where we use the fact that $[\mathbf{r} \cdot \mathbf{E}(t), \mathbf{r}] = 0$. Thus, the velocity operator in the length gauge is independent of perturbation.

On the other hand, in the velocity gauge, the total Hamiltonian is written as

$$H_{\text{tot}}^V(\mathbf{k}, t) = H(\mathbf{k}, t)|_{\mathbf{k}\to\mathbf{k}+\mathbf{A}(t)}.$$

Thus, we can write

$$\mathbf{v}^A = \frac{d\mathbf{r}}{dt} = i\left[H_{\text{tot}}^V(\mathbf{k}, t), \mathbf{r}\right] = i\left[H(\mathbf{k}, t)|_{\mathbf{k}\to\mathbf{k}+\mathbf{A}(t)}, \mathbf{r}\right]$$
$$= i[\mathbf{k}, \mathbf{r}] \partial_\mathbf{k}\left(H(\mathbf{k}, t)|_{\mathbf{k}\to\mathbf{k}+\mathbf{A}(t)}\right) = \partial_\mathbf{k}\left(H(\mathbf{k}, t)|_{\mathbf{k}\to\mathbf{k}+\mathbf{A}(t)}\right). \tag{B.3}$$

So, the velocity operator in the velocity gauge depends on the external perturbation.

# C Obtaining length gauge conductivities from velocity gauge conductivities

Now, we demonstrate how to switch between the conductivity formulas of the velocity gauge and the length gauge. In the velocity gauge conductivity formulas, there are matrix elements of various order derivatives of the Hamiltonian, which we need to express in terms of the

matrix elements of the first-order derivative of the Hamiltonian. The matrix elements of the second-order derivative can be written as

$$h_{\alpha\beta}^{xz} = \partial_z h_{\alpha\beta}^x - i[\xi^z, h^x]_{\alpha\beta}, \tag{C.1}$$

where we have defined $\xi_{\alpha\gamma}^z \equiv i\langle\phi_\alpha(t)|\partial_{k_z}\phi_\gamma(t)\rangle$ as a generalize Berry connection. The proof of the above equation can be written readily:

$$
\begin{aligned}
h_{\alpha\beta}^{xz} &= \langle\phi_\alpha(t)|\partial_{k_z}\partial_{k_x}H|\phi_\beta(t)\rangle = \partial_{k_z}\langle\phi_\alpha(t)|\partial_{k_x}H|\phi_\beta(t)\rangle - I, \\
I &= \langle\partial_{k_z}\phi_\alpha(t)|\partial_{k_x}H|\phi_\beta(t)\rangle + \langle\phi_\alpha(t)|\partial_{k_x}H|\partial_{k_z}\phi_\beta(t)\rangle \\
&= -i\sum_\gamma\left(-\xi_{\alpha\gamma}^z h_{\gamma\beta}^x + h_{\alpha\gamma}^x \xi_{\gamma\beta}^z\right) = i[\xi^z, h^x]_{\alpha\beta},
\end{aligned}
\tag{C.2}
$$

where for obtaining the third line we inserted the complete set $\mathbf{1} = \sum_\gamma |\phi_\gamma(t)\rangle\langle\phi_\gamma(t)|$ and also used the fact that $\langle\partial_{k_z}\phi_\alpha(t)|\phi_\gamma(t)\rangle + \langle\phi_\alpha(t)|\partial_{k_z}\phi_\gamma(t)\rangle = \partial_{k_z}\delta_{\alpha\gamma} = 0$.

It is an easy task to calculate the Fourier components of Eq. (C.1) as follows

$$h_{\alpha\beta}^{xz(n)} = \partial_{k_z}h_{\alpha\beta}^{x(n)} - i\sum_{\gamma j}\left(\xi_{\alpha\gamma}^{z(j+n)}h_{\gamma\beta}^{x(-j)} - h_{\alpha\gamma}^{x(j+n)}\xi_{\gamma\beta}^{z(-j)}\right). \tag{C.3}$$

Using Eq. (C.3), we can derive the following relations::

$$
\begin{aligned}
\sum_{\alpha\mathbf{k}} f_\alpha h_{\alpha\alpha}^{xz(n)} &= \sum_{\alpha\mathbf{k}}\left[f_\alpha\partial_{k_z}h_{\alpha\alpha}^{x(n)} - if_\alpha\sum_{\beta j}\left(\xi_{\alpha\beta}^{z(j+n)}h_{\beta\alpha}^{x(-j)} - h_{\alpha\beta}^{x(j+n)}\xi_{\beta\alpha}^{z(-j)}\right)\right] \\
&= \sum_{\alpha\mathbf{k}}\left[f_\alpha\partial_{k_z}h_{\alpha\alpha}^{x(n)} - if_\alpha\sum_{\beta j}\left(\xi_{\alpha\beta}^{z(-j)}h_{\beta\alpha}^{x(j+n)} - h_{\alpha\beta}^{x(j+n)}\xi_{\beta\alpha}^{z(-j)}\right)\right] \\
&= \sum_{\alpha\mathbf{k}}\left[-h_{\alpha\alpha}^{x(n)}\partial_{k_z}f_\alpha + i\sum_{\beta j}f_{\beta\alpha}h_{\beta\alpha}^{x(j+n)}\xi_{\alpha\beta}^{z(-j)}\right] \\
&= \sum_{\alpha\mathbf{k}}\left[-h_{\alpha\alpha}^{x(n)}\partial_{k_z}f_\alpha + \sum_{\beta j}\frac{f_{\beta\alpha}h_{\beta\alpha}^{x(j+n)}h_{\alpha\beta}^{z(-j)}}{\epsilon_{\alpha\beta}+j\Omega}\right],
\end{aligned}
\tag{C.4}
$$

where in the third line we performed integration by parts, taking advantage of the Brillouin zone being a closed path, and in the final line we applied the relationship between matrix elements of the position and velocity operators:

$$\xi_{\alpha\neq\beta}^{z(-j)} = \frac{-i}{\epsilon_{\alpha\beta}+j\Omega}h_{\alpha\beta}^{z(-j)}, \tag{C.5}$$

that was mentioned in Ref. [36]. Inserting Eq. (C.4) in the first term of Eq. (4) eliminates the divergence term and yields the length-gauge expression

$$\sigma_{xz}^{[1](n)}(\omega_1) = \sum_{\alpha\mathbf{k}}h_{\alpha\alpha}^{x(n)}\frac{1}{i\omega_1}\partial_{k_z}f_\alpha - i\sum_{j\alpha\beta\mathbf{k}}f_{\beta\alpha}\frac{h_{\beta\alpha}^{x(j+n)}h_{\alpha\beta}^{z(-j)}}{(\epsilon_{\alpha\beta}+j\Omega-\omega_1)(\epsilon_{\alpha\beta}+j\Omega)}. \tag{C.6}$$

The first term represents the intraband contribution, while the second term corresponds to the interband contribution. The only divergence at zero frequency occurs in the intraband term when $\partial_{k_z}f_\alpha \neq 0$, which arises in partially filled Floquet bands. This divergence is physical and

analogous to the Drude peak in metals, with a key distinction: here, a DC field may induce a divergent AC response [36].

To determine the second-order response in the length gauge, we need to express the second-order and third-order derivatives of the Hamiltonian (as in Eq. (C.3)) in terms of its first derivative in the formula (5). The matrix elements of the third-order derivative of the Hamiltonian can be related to the second-order derivative in the following manner:

$$h^{xyz}_{\alpha\beta} = \partial_z h^{xy}_{\alpha\beta} - i[\xi^z, h^{xy}]_{\alpha\beta}. \tag{C.7}$$

# D  Third order conductivity

For the sake of completeness, we determine the third-order response of Floquet systems in this section. First, we evaluate the third-order density matrix $\rho^{[3]}_{\alpha\beta}$ using Eq. (3) from the main text and Eqs. (A.11), and (A.10). We introduce some shorthand notations $\epsilon^{(j)}_{\alpha\beta} \equiv \epsilon_\alpha - \epsilon_\beta + j\Omega$ and $\omega_1 + \omega_2 \equiv \omega_{12}$, $\omega_1 + \omega_2 + \omega_3 \equiv \omega$. The non-symmetrized third-order conductivity at frequency of $\omega_1 + \omega_2 + \omega_3 + n\Omega$ is composed of eight terms which can be written as

$$
\sigma^{[3](n)}_{wxyz}(\omega_1, \omega_2, \omega_3) \tag{D.1}
$$

$$
= \frac{-i}{\omega_1\omega_2\omega_3}\left\{\sum_\alpha \frac{h^{wxyz(n)}_{\alpha\alpha}}{3!}f_\alpha + \sum_{\alpha\beta j}\frac{\frac{1}{2}f_{\beta\alpha}h^{wxy(j+n)}_{\beta\alpha}h^{z(-j)}_{\alpha\beta}}{\omega_1 - \epsilon^{(j)}_{\alpha\beta}} + \frac{\frac{1}{2}f_{\beta\alpha}h^{wx(j+n)}_{\beta\alpha}h^{yz(-j)}_{\alpha\beta}}{\omega_{12} - \epsilon^{(j)}_{\alpha\beta}}\right.
$$

$$
+ \frac{\frac{1}{3!}f_{\beta\alpha}h^{w(j+n)}_{\beta\alpha}h^{xyz(-j)}_{\alpha\beta}}{\omega - \epsilon^{(j)}_{\alpha\beta}}
$$

$$
+ \sum_{\alpha\beta\gamma j_1 j_2}h^{wx(j_1+j_2+n)}_{\beta\alpha}h^{y(-j_2)}_{\alpha\gamma}h^{z(-j_1)}_{\gamma\beta}I(\omega_1, \omega_2) + \frac{1}{2}h^{w(j_1+j_2+n)}_{\beta\alpha}h^{xy(-j_2)}_{\alpha\gamma}h^{z(-j_1)}_{\gamma\beta}I(\omega_1, \omega_{23})
$$

$$
+ \frac{1}{2}h^{w(j_1+j_2+n)}_{\beta\alpha}h^{x(-j_2)}_{\alpha\gamma}h^{yz(-j_1)}_{\gamma\beta}I(\omega_{12}, \omega_3)
$$

$$
+ \sum_{\alpha\beta\gamma\theta j_1 j_2 j_3}\frac{h^{w(j_1+j_2+j_3+n)}_{\beta\alpha}h^{x(-j_3)}_{\alpha\theta}h^{y(-j_2)}_{\theta\gamma}h^{z(-j_1)}_{\gamma\beta}}{\left(\epsilon^{(j_1+j_2+j_3)}_{\alpha\beta} - \omega\right)\left(\epsilon^{(j_1+j_2)}_{\theta\beta} - \omega_{12}\right)}\left[\frac{f_{\beta\gamma}}{\epsilon^{(j_1)}_{\gamma\beta} - \omega_1} - \frac{f_{\gamma\alpha}}{\epsilon^{(j_2)}_{\theta\gamma} - \omega_2}\right]
$$

$$
\left. - \frac{h^{w(j_1+j_2+j_3+n)}_{\beta\alpha}h^{x(-j_3)}_{\theta\beta}h^{y(-j_2)}_{\alpha\gamma}h^{z(-j_1)}_{\gamma\theta}}{\left(\epsilon^{(j_1+j_2+j_3)}_{\alpha\beta} - \omega\right)\left(\epsilon^{(j_1+j_2)}_{\alpha\theta} - \omega_{12}\right)}\left[\frac{f_{\theta\gamma}}{\epsilon^{(j_1)}_{\gamma\theta} - \omega_1} - \frac{f_{\gamma\alpha}}{\epsilon^{(j_2)}_{\alpha\gamma} - \omega_2}\right]\right\}.
$$

The above results should be symmetrized concerning the interchange of $(\omega_1, z) \longleftrightarrow (\omega_2, y) \longleftrightarrow (\omega_3, x)$ to yield the final optical conductivity.

# E  Nonlinear response of driven SSH model

Here, we would like to utilize the formalism developed in this paper to compute the linear and nonlinear response of the driven Su-Schrieffer-Heeger (SSH) model in the velocity gauge, and we compare these results with those obtained in the length gauge. The SSH model is a one-dimensional system defined on a bipartite lattice, originally introduced to describe electronic states in polyacetylene [52]. The Hamiltonian we analyze can be expressed in momentum space as follows:

$$H(k, t) = \frac{\Omega}{2}[(0.5 + \cos k)\sigma_z - \sin k\sigma_y] + A\cos(\Omega t)\sigma_x. \tag{E.1}$$

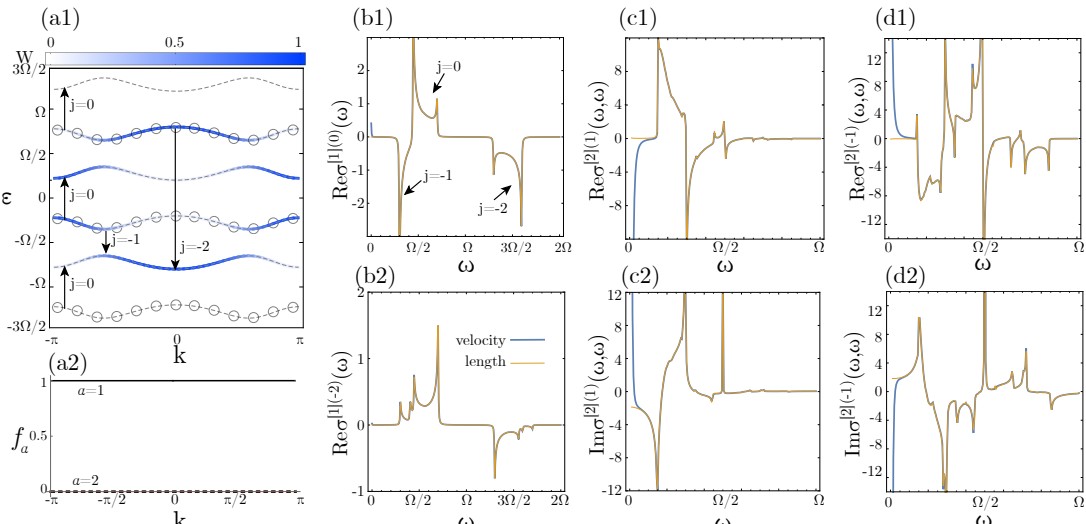

Figure 5: (Color online) (a1) Quasienergy bands, (a2) ideal occupation of Floquet states, (b) first-order optical response of a driven SSH model to a probe field at a frequency of $\omega$ using Eqs. (4) and (E.1). (c), (d) finite elements of the second-order conductivity of the driven SSH model in the velocity gauge (blue line) and length gauge (orange line). The circles show a band filled, and the color scale shows the physical weight defined in the main text. Arrows indicate the $j$-photon-assisted optical transitions in (a1). The parameters are $\Omega = 1, A = 0.3$.

By applying Eqs. (4) and (5), the nonlinear response in the velocity gauge can be readily calculated. For the length gauge results, we use Eqs. (5), (6), and (8), along with Eq. (c2) from Ref. [36].

In Fig. (5)(a1), the quasienergy band structure is displayed, while (a2) depicts the ideal occupation of Floquet states, where the lower Floquet band is fully occupied and the upper Floquet band is entirely empty. The linear and nonlinear responses are shown in Figs. (5)(b) and (c),(d), respectively, demonstrating excellent agreement between the velocity and length gauges. As established in this paper, the length and velocity gauges are indeed equivalent, a fact confirmed by the numerical results. The only distinction lies in the spurious divergence observed in the zero-frequency limit within the velocity gauge, which disappears in the length gauge. However, when the bands are partially filled, intraband components can become active in the length gauge, leading to a genuine divergence at the zero-frequency limit, known as the Drude peak [36].

Finally, we address a key consideration for nonlinear response calculations in the length gauge. As discussed in Ref. [36], selecting a smooth gauge for the Bloch wave function phases across the Brillouin zone is essential, particularly for the intraband-interband component $\sigma^{[2]ie}$. In our numerical implementation, we ensure gauge smoothness through manual inspection and verify result convergence. Cross-checking the Berry connection matrix elements against the velocity matrix elements (Eq. (C.5)) provides additional validation.

An alternative approach circumvents direct computation of the Berry connection and covariant derivative by employing sum rules derived from the fundamental relation (Eq. (C.1)) which can be reformulated as follows:

$$ih_{\alpha\beta}^{xz} = i\left(h_{\alpha\beta}^{x}\right)_{;k_z} + [\mathbf{r}_e^z, h^x]_{\alpha\beta}, \tag{E.2}$$

where $\mathbf{r}_e^z$ denotes the interband position operator, with matrix elements given by $\mathbf{r}_{\alpha\beta}^e = \delta_{\alpha\beta}\xi_{\alpha\beta}^z$. Using Eq. (E.2) and (C.5), we can express the covariant derivative in terms of velocity oper-

ator matrix elements, thereby eliminating the need to compute derivatives of the Bloch wave functions directly.

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
