# Peer review of "Velocity gauge formulation of nonlinear optical response in Floquet quantum systems"

_SciPost Physics Core, doi:SciPost Phys. Core 8, 049 (2025)_

## Round 1 · Referee Report · Anonymous (Referee 1) · 2025-5-22

Strengths

Derive new formula for linear and non-linear conductivity of quantum floquet system in velocity gauge. Derivations are clear, results correct.

Weaknesses

Small points to be clarified, see report.

Report

In this manuscript, the authors study the linear and nonlinear conductivity of a Floquet quantum system using a velocity gauge. They derived formulas for first-, second-, and third-order conductivity and compared their results with those of the length gauge approach. I found the manuscript to be both correct and interesting. However, there are a few points that the authors should discuss or clarify before suggesting publication:

1) Is the Drude term they obtained real or an artifact? Many papers discuss the application of sum rules to remove unrealistic divergences in the velocity gauge, including the Drude term for insulators. See, for instance, sec. II.2 and II.3 of PRB 95, 155203 or PRB 48, 11705. Do these sum rules also apply in the Floquet case?

2) On page 5, the sentence "It is important to note that the velocity gauge..." is unclear. Are the two gauges not equivalent? Could the authors clarify this point?

3) On page 7, when the authors speak of "spurious" divergences that cancel, etc., they could add a reference.

4) In Eq. 4 and following, they could explicitly write the dependence of σ(ω₁). This would make the equation clearer. The same goes for Eq. 5, σ(ω₁, ω₂).

5) On page 7, the second response they calculate is the "sum frequency generation" ω = ω₁ + ω₂, am I right? They should specify this in the text.

6) On page 9, when they speak of "moving to a rotating frame," this is simply the gauge transformation (see Sec. III of PRA 79, 053415) or not?

7) On page 9, in section III.A, the authors use a gauge for the Hamiltonian and a gauge to calculate the response. Is this allowed? Is it correct?

8) I have a comment regarding the imaginary part that is added to the energies. In principle, adding an imaginary part to the final formula for conductivity is correct only in the length gauge. In the velocity gauge, however, this introduces a small error. This has been demonstrated, for example, in PRB A 79, 053415 or PRB A 36, 2763. The error introduced is probably small, but I wanted to point out to the authors that the procedure is not so straightforward.

9) In Appendix C, when discussing formulas in the length gauge, how do they calculate the generalized Berry connection? Don't they have a problem with the phase of the wave function at different k-points?

Requested changes

see report

Recommendation

Ask for minor revision

  • validity: good
  • significance: good
  • originality: good
  • clarity: good
  • formatting: good
  • grammar: good

Author:  S. Sajad Dabiri  on 2025-06-12  [id 5562]

(in reply to Report 1 on 2025-05-22)

The referee writes:

In this manuscript, the authors study the linear and nonlinear conductivity of a Floquet quantum system using a velocity gauge. They derived formulas for first-, second-, and third-order conductivity and compared their results with those of the length gauge approach. I found the manuscript to be both correct and interesting. However, there are a few points that the authors should discuss or clarify before suggesting publication:

Our response: We appreciate the referee's positive opinion about the manuscript.

The referee writes:

1) Is the Drude term they obtained real or an artifact? Many papers discuss the application of sum rules to remove unrealistic divergences in the velocity gauge, including the Drude term for insulators. See, for instance, sec. II.2 and II.3 of PRB 95, 155203 or PRB 48, 11705. Do these sum rules also apply in the Floquet case?

Our response: We thank the referee for raising this question. We might say the divergences in the velocity gauge are not always physical at low frequencies. For example, when the Floquet bands are either filled or empty, there should not be any Drude peaks at low frequencies. However, it would be beneficial to use the length gauge according to (dabiri2025dynamical) at low frequencies. There are generalized sum rules that relate the velocity and length gauges. The key identity for this is given by Eq.~(C3) in the main text, which implies: Eq.(C4) of new manuscript. $\sum\limits_{\alpha \mathbf{k}}{{f_\alpha }h_{\alpha \alpha }^{xz(n)}}$ $=\sum\limits_{\alpha \mathbf{k}} [f_\alpha \partial_{k_z} h_{\alpha \alpha }^{x(n)}-i{f_\alpha}\sum\limits_{\beta j} (\xi_{\alpha \beta }^{z(j+n)} h_{\beta \alpha}^{x(-j)}- h_{\alpha \beta}^{x(j+n)}\xi_{\beta \alpha}^{z(-j)})]$ $ =\sum\limits_{\alpha \mathbf{k}} [f_\alpha \partial_{k_z} h_{\alpha \alpha }^{x(n)}-i{f_\alpha}\sum\limits_{\beta j} (\xi_{\alpha \beta }^{z(-j)} h_{\beta \alpha}^{x(j+n)}- h_{\alpha \beta}^{x(j+n)}\xi_{\beta \alpha}^{z(-j)})]$ $ =\sum\limits_{\alpha \mathbf{k}} [ -h_{\alpha \alpha }^{x(n)}\partial_{k_z} f_\alpha +i \sum\limits_{\beta j} f_{\beta\alpha} h_{\beta \alpha}^{x(j+n)}\xi_{\alpha \beta }^{z(-j)} ]$ $ =\sum\limits_{\alpha \mathbf{k}} [ -h_{\alpha \alpha }^{x(n)}\partial_{k_z} f_\alpha + \sum\limits_{\beta j} \frac{f_{\beta\alpha} h_{\beta \alpha}^{x(j+n)}\xi_{\alpha \beta }^{z(-j)}}{\epsilon_{\alpha\beta}+j\Omega} ]$ where in the third line we perform integration by parts, taking advantage of the Brillouin zone being a closed path, and in the final line we also apply the relationship between matrix elements of the position and velocity operators: $\xi_{\alpha \ne \beta}^{z(-j)}=\frac{-i}{\epsilon_{\alpha \beta }+j\Omega} h_{\alpha \beta }^{z(-j)}$ that was mentioned in Ref.(dabiri2025dynamical). Inserting above equation in the first term of Eq.~(4) in the main text leads to the cancellation of the divergence term and yields the length-gauge expression $\sigma_{xz}^{[1]}(\omega_1)=\sum\limits_{\alpha \mathbf{k}} h_{\alpha \alpha }^{x(n)} \frac{1}{i\omega_1}\partial_{k_z} f_\alpha-i \sum\limits_{j\alpha \beta \mathbf{k}}f_{\beta\alpha} \frac{f_{\beta\alpha} h_{\beta \alpha}^{x(j+n)}\xi_{\alpha \beta }^{z(-j)}}{(\epsilon_{\alpha\beta}+j\Omega-\omega_1)(\epsilon_{\alpha\beta}+j\Omega)} $ The first term represents the intraband contribution, while the second term corresponds to the interband contribution. The only divergence at zero frequency occurs in the intraband term when $\partial_{k_z} f_\alpha\neq 0$, which arises in partially filled Floquet bands. This divergence is physical and analogous to the Drude peak in metals, with a key distinction: here, a DC field may induce a divergent AC response.

The referee writes:

2) On page 5, the sentence "It is important to note that the velocity gauge..." is unclear. Are the two gauges not equivalent? Could the authors clarify this point?

Our response: Thank you for bringing this to our attention. What we want to emphasize is how and where we can practically use those gauges. The two gauges provide the same results; however, due to some practical challenges, one gauge works well for a certain problem rather than the other. The velocity gauge, where the vector potential is used to obtain the electric field, is especially useful in high frequency THz (terahertz) and strong-field physics because it emphasizes kinetic energy contributions. While length gauge may require larger basis sets to capture high-energy transitions accurately. The length gauge is generally simpler when dealing with bound states and many-body systems, like atoms, molecules, or finite systems where dipole moment is a well defined operator. Moreover, the length gauge is more suitable for lower-order perturbation responses as the higher orders involve complicated matrix elements.

The referee writes:

3) On page 7, when the authors speak of "spurious" divergences that cancel, etc., they could add a reference.

Our response: Thank you for pointing out that issue. The issue is fixed in the new version of the manuscript.

The referee writes:

4) In Eq. 4 and following, they could explicitly write the dependence of $\sigma(\omega_1)$. This would make the equation clearer. The same goes for Eq. 5, $\sigma(\omega_1,\omega_2)$

Our response: Thank you for pointing out that issue. The issue is fixed in the new version of the manuscript.

The referee writes:

5) On page 7, the second response they calculate is the "sum frequency generation" $\omega=\omega_1+\omega_2$, am I right? They should specify this in the text.

Our response: Thanks for reminding us of this point. We emphasized that Eq.~(5) shows the response at the frequency of $\omega=\omega_1+\omega_2+n\Omega$.

The referee writes:

6) On page 9, when they speak of "moving to a rotating frame," this is simply the gauge transformation (see Sec. III of PRA 79, 053415) or not?

Our response: Thank you for asking this question. Yes, this is a simple gauge transformation. Please see Eqs.~(4-7) Ref.ventura2017gauge where they discuss that such a transformation between two gauges shows their equivalence. One can explain the gauge transformation as follows. It is known that the Schrodinger equation in the presence of an electromagnetic field can be written as $i\partial_t\psi =H(-i\nabla +\mathbf{A})\psi -\phi \psi $ Where $\psi =\psi (\mathbf{r},t),\mathbf{A}=\mathbf{A}(\mathbf{r},t),\phi =\phi (\mathbf{r},t)$ are wave function, vector potential and scalar potential. A gauge transformation is obtained through the substitution $\mathbf{A}^\prime (\mathbf{r},t)=\mathbf{A}(\mathbf{r},t)+\nabla \Lambda (\mathbf{r},t)$, $\phi^\prime(\mathbf{r},t)=\phi (\mathbf{r},t)-{\partial_t}\Lambda (\mathbf{r},t).$ Then, the new wave function is $\psi^\prime(\mathbf{r},t)=e^{-i\Lambda (\mathbf{r},t)}\psi (\mathbf{r},t).$ As we explain in the answer to the next question, in our problem we choose $\Lambda (\mathbf{r},t)=\mathbf{r}.\mathbf{A}(t)$. So, this is indeed a gauge transformation.

The referee writes:

7) On page 9, in section III.A, the authors use a gauge for the Hamiltonian and a gauge to calculate the response. Is this allowed? Is it correct?

Our response: Thank you for asking this question. Yes, that is right. Please assume, for example, we use Eq.~(9) instead of Eq.~(8) of the main text and find out the optical conductivity using $\mathcal{H}^\prime(k,t)$. As we have shown in the main text, two Hamiltonians are related to each other in different gauges through a time-dependent unitary transformation. According to Eq.~(10) of the main text, if $|\phi_\alpha(t)\rangle$ is a Floquet quasi-mode of $\mathcal{H}$, then $|\phi^\prime_\alpha(t)\rangle=U_R^\dagger(t)|\phi_\alpha(t)\rangle$ is the Floquet quasi-modes of $\mathcal{H}^\prime$ with the same quasi-energy. Because of the form of matrix elements used in our formulation (See Eqs.( 4), (5) and (D1) of the main text), this unitary transformation does not change the final results, for example, we have: $ h_{\alpha \beta}^{\prime i}=\langle \phi_\alpha^\prime(t)|\partial_{k_i}\mathcal{H}^\prime(k,t)|\phi_\beta^\prime(t)\rangle =\langle \phi_\alpha(t)|U_R(t)U_R^\dagger(t)\partial_{k_i}\mathcal{H}(k,t)U_R(t)U_R^\dagger(t)|\phi_\beta(t)\rangle =h_{\alpha \beta }^i$ where we have used the fact that $U_R(t)$ is independent of momentum, so $\partial_k [U_R^\dagger(t)\cdot\partial_t U_R(t)]=0.$ We include this information in the main text.

The referee writes:

8) I have a comment regarding the imaginary part that is added to the energies. In principle, adding an imaginary part to the final formula for conductivity is correct only in the length gauge. In the velocity gauge, however, this introduces a small error. This has been demonstrated, for example, in PRB A 79, 053415 or PRB A 36, 2763. The error introduced is probably small, but I wanted to point out to the authors that the procedure is not so straightforward.

Our response: Thanks for pointing this out. We tried to replace $\omega$ with $\sqrt{\omega(\omega+i\eta)}$ as suggested in PRB 95, 155203, and got slightly better results. So, we replotted all the figures. We believe one should use the sum rules obtained from Eqs.~(C1-C7) and (E2) of the new version of the manuscript to obtain better results at low frequencies. The length gauge results are more accurate than velocity gauge results, especially at low frequencies; nevertheless, the analytical results are the same, which is shown in Eq.~(C1)-(C7). Future studies can be carried out using more realistic broadening functions, as the Referee correctly pointed out.

The referee writes:

9) In Appendix C, when discussing formulas in the length gauge, how do they calculate the generalized Berry connection? Don't they have a problem with the phase of the wave function at different k-points?

Our response: We appreciate the Referee for asking this question. Appendix C shows how to obtain the exact length gauge formulas from velocity gauge formulas. Numerical calculations in the length gauge have been carried out according to the formulas in Ref.~(\cite{dabiri2025dynamical}). There, we have noted that choosing a smooth gauge for the phase of Bloch wave functions across the Brillouin zone is needed, especially for the intraband-interband part $\sigma^{[2]ie}$. We have taken a smooth gauge in our numerical calculations by inspection and checked that the results are convergent. Comparing the Berry connection matrix elements and velocity matrix elements $\xi_{\alpha \ne \beta}^{z(-j)}=\frac{-i}{\epsilon_{\alpha \beta}+j\Omega}h_{\alpha \beta }^{z(-j)}$ is a way to verify the results. The other way to avoid calculating the Berry connection and covariant derivative is by using the sum rules, which can be derived from the key relation (C1) in the main text. One can rewrite (C1) as: $ih_{\alpha \beta}^{xz}=i( h_{\alpha \beta}^x )_{;k_z}+ [\mathbf{r}_e^z, h^x]$ $_{\alpha \beta}.$ where $\mathbf{r}_e^z$ is the interband position operator whose matrix elements are $\mathbf{r}^e_{\alpha\beta}=\delta_{\alpha\beta}\xi^z_{\alpha\beta}$ and we know that $\xi_{\alpha \ne \beta}^{z(-j)}=\frac{-i}{\epsilon_{\alpha \beta}+j\Omega}h_{\alpha \beta }^{z(-j)}$. From the above relation, one can derive the covariant derivative in terms of matrix elements of the velocity operator, which does not involve the derivative of Bloch wave functions.

---

## Round 2 · Author Response

We thank you for sending us the report from the referees. We also thank the referee for his\her positive evaluation of our manuscript. We would like to inform you that we have incorporated all the constructive suggestions and have replied to the comments, which have improved our manuscript.

---

## Round 2 · List of Changes

1 The figures are refreshed, showing better results in velocity gauge at low frequencies by using other replacement $\omega \rightarrow \sqrt{\omega(\omega+i\eta)}$ method according to the referee's reply. 2 It is proved that two Rabi Hamiltonians in different gauges yield the same optical conductivity. 3 We have shown how one can obtain sum-rules and, specifically, the linear response in the length gauge is derived from the velocity gauge formula. 4 We have shown how to avoid the derivative of the Bloch wave functions in calculations using sum rules. 5 Some points are added to the text according to the referee's suggestions for more clarity.

---

## Editorial Decision

published